# Topology mediates transport of nanoparticles in macromolecular networks

Xiaobin Dai[1], Xuanyu Zhang[1], Lijuan Gao[1], Ziyang Xu[1] & Li-Tang Yan [1✉]

Diffusion transport of nanoparticles in confined environments of macromolecular networks is common in diverse physical systems and regulates many biological responses. Macromolecular networks possess various topologies, featured by different numbers of degrees and genera. Although the network topologies can be manipulated from a molecular level, how the topology impacts the transport of nanoparticles in macromolecular networks remains unexplored. Here, we develop theoretical approaches combined with simulations to study nanoparticle transport in a model system consisting of network cells with defined topologies. We find that the topology of network cells has a profound effect on the free energy landscape experienced by a nanoparticle in the network cells, exhibiting various scaling laws dictated by the topology. Furthermore, the examination of the impact of cell topology on the detailed behavior of nanoparticle dynamics leads to different dynamical regimes that go beyond the particulars regarding the local network loop. The results might alter the conventional picture of the physical origin of transport in networks.

[1] State Key Laboratory of Chemical Engineering, Department of Chemical Engineering, Tsinghua University, Beijing 100084, China. ✉email: ltyan@mail.tsinghua.edu.cn

Topology is one of the most important concepts in modern physics[1,2]. It plays a critical role in dictating the properties of many materials, e.g., topological electronic materials[3,4], mesoscopic photonic materials[5,6], and macroscopic mechanical lattices[7,8], etc. Much of our knowledge about the topology-mediated behaviors is based on the strong correlation materials where the interaction energy dominates the kinetic energy in controlling the particle transport[9]. However, the topological effects on the particle transport in soft materials, where entropy contribution can overwhelm the interaction energy, have been scarcely understood.

On the other hand, macromolecular networks are among the most universal structural bases of soft matter systems, ranging from commodity materials[10–12], such as elastomers, gels, and soft actuators, to biological materials[13–15], such as the extracellular matrix, mucus, and tumor tissues. As a consequence, diffusion transport of nanoparticles in confined environments of macromolecular networks is a fundamental problem underlying many important physical processes and biological responses, for example, from purification in porous materials[16] to pathogen infection[10,17,18], drug release[19,20], and human evacuation behavior[21]. One critical factor dictating network properties is the topological structure[22–24], which can be characterized by degree $n$, functionality $k$ and genus $g$, as schematically shown in Fig. 1a. The past decades have witnessed substantial progress in the understanding and manipulation of topological structures of macromolecular networks, resulting in well-controlled and even programmable network topologies[25–27]. Although the topology of networks and defects has been demonstrated to be important in many phenomena and theories[22,23], topological aspects of nanoparticle transport dynamics remain unexplored.

The classical pictures of the dynamics of nanoparticles confined in macromolecular networks have mostly been built on the free energy barrier of the local network loop[12,28,29]. In contrast, elucidating the physical origin of topological effects requires a full view of the free energy landscape sculpted by the topology and thereby its impact on the nanoparticle dynamics[30,31], which, however, has thus far been lacking. Experiments do not yet have the resolution to detect the free energy landscape experienced by a nanoparticle in a macromolecular network. Therefore, theoretical approaches that explicitly quantify the free energy landscape around the particle and explore the consequences for the dynamic behavior play a vital role.

To understand the nature of topological effects on particle transport dynamics, here, we develop a theoretical framework to provide a rigorous analysis of the relation of the free energy landscape and diffusive dynamics to the topological structure for a particle in network cells of permanently cross-linked macromolecular networks. In combination with simulations, we demonstrate and explain the profound effects of network topology on particle diffusion. The theoretical models reveal distinct scaling regimes regarding the free energy landscape and particle dynamics, dictated by the topology.

## Results

**Network topology and free energy landscape.** We first develop a theoretical model by coupling the particle effect into the theory of macromolecular-network elasticity to determine the free energy

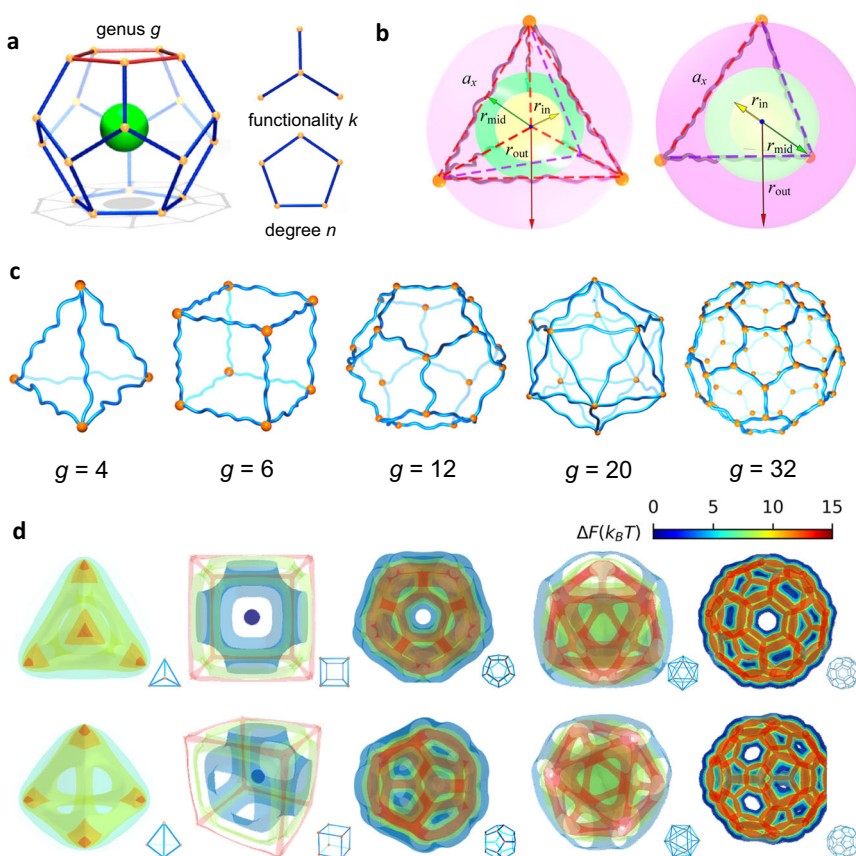

**Fig. 1 Detailed overview of network topology and the free energy landscape. a** Schematic representation of a particle in a macromolecular network with topological parameters: genus $g$, degree $n$, and functionality $k$. **b** Schematic diagram of the scaling parameters of a network cell at $g = 4$, where arrows shown in yellow, green and red denote respectively inradius $r_{in}$, midradius $r_{mid}$, and circumradius $r_{out}$ of the cell from different views. **c** Schematic representation of macromolecular network cells with different topologies. **d** Isosurfaces of the free energy change $\Delta F$ of a particle in networks with different topologies marked at the right bottom, where the scaled diameter $d/a_x = 1.4$. The color bar on the top right corner encodes the value of $\Delta F$.

landscape experienced by the nanoparticle. Full technical details on the simulation model are described in the Methods and Supplementary Information I. To establish clear quantitative trends, we propose a way to examine the topological effects by designing topology network cells that resemble a series of Platonic or Archimedean polyhedra, in which the strands and cross-links of the elementary network constitute the edges, with average mesh size $a_x$ defined as the root-mean-square end-to-end distance of the strands, and vertices of polyhedra (Fig. 1b, c). Given that these polyhedral have regular geometries and are homeomorphic to spheres, the topologies of corresponding network cells can be simply determined by $g$. Figure 1c shows the representative network cells with increasing $g$. Furthermore, as all Platonic or Archimedean polyhedra have an inscribed sphere tangent to the faces, a midsphere tangent to the edges, and a circumscribed sphere through the vertices (Fig. 1b), the normalized radii of these spheres, i.e., $r_{in}/a_x$, $r_{mid}/a_x$ and $r_{out}/a_x$, allow scaling parameters to characterize the cell topologies and can be given based on $g$ and $n$,

$$\frac{r_{in}}{a_x} = \frac{1}{2} \cot\left(\frac{\pi}{n}\right) \tan\left(\frac{\theta}{2}\right)$$
$$\frac{r_{mid}}{a_x} = \frac{1}{2} \cot\left(\frac{\pi}{n}\right) \sec\left(\frac{\theta}{2}\right) \quad (1)$$
$$\frac{r_{out}}{a_x} = \frac{1}{2} \cot\left(\frac{\pi}{n}\right) \sqrt{\sec^2\left(\frac{\theta}{2}\right) + \tan^2\left(\frac{\pi}{n}\right)}$$

where $\theta$ represents the dihedral angle between any two faces.

Figure 1d and Supplementary Fig. S1 show constant free energy isosurfaces for a nanoparticle, with diameter $d/a_x = 1.4$, in some representative network cells corresponding to Fig. 1c. Despite the same sizes of both the nanoparticle and network mesh, the different topologies of these cells dramatically change the free energy landscape experienced by the nanoparticle. Depending on $g$, the free energy landscape exhibits symmetry but is obviously anisotropic, which suggests diverse free energy barriers for nanoparticle diffusion across different cells, in stark contrast to the certain free energy barrier of the local network loop[12,28,29]. Nevertheless, disregarding the cell topology, the free energy is lowest in the core region of each cell, and there is a local free energy minimum at the center of each face; the connection of the two free energy minima yields the minimum energy path (MEP)[32] for the transition of a nanoparticle from the cell to its neighbor (see Supplementary Information II for more information).

**Topology-dictated scaling regimes of free energy change.** To delineate the free energy experienced by a nanoparticle in a network cell, we examine the MEP for nanoparticles with various sizes at $g = 6$. For this purpose, one MEP is chosen as the $z$ axis with its origin positioned at the cell center (see the inset of Fig. 2a). Figure 2a shows the profiles of the free energy change, $\Delta F(z) = F(z) - F(0)$, with the zero-point free energy $F(0)$, upon increasing $z$ from the origin to the face center for various $d$ in the log-log scale. To corroborate our theoretical results, we also perform Monte Carlo (MC) simulations, and the MC results are compared to the theoretical results for some representative particle sizes in Fig. 2b, where the standard error is estimated to be within $0.3k_BT$, with the Boltzmann constant $k_B$ and the temperature $T$. The good agreement between the simulated and theoretical results indicates that the theoretical model faithfully captures the free energy change experienced by the nanoparticle entrapped in network cells.

A close examination of the profiles in Fig. 2a, b leads to an intriguing observation: all the path dependences of $\Delta F(z)$ can be classified into four regimes based on the particle size, as approximately bounded by shaded sections. For a large nanoparticle, $\Delta F(z)$ exhibits a power law dependence on $z$, $\Delta F \sim z^\eta$, where $\eta$ is the scaling exponent and $\eta = 1$ in this regime. Upon reducing the nanoparticle size, in the following regime, $\eta$ crosses over from 1

to 2. The crossover behavior becomes more evident for a larger $d$. With a further reduction of $d$, the profiles become nonconsecutive, in which $\Delta F(z)$ initially remains zero but then abruptly jumps to a large value. Such an abrupt jump reminds us of the obstruction effect of diffusion in a polymer gel network, as reported in experiments[33]. When $d$ is sufficiently small, the effect of the network cell is trivial such that $\Delta F(z)$ remains zero throughout the path in the last regime. For a legible presentation of the above four scaling regimes, we systematically compute the $\Delta F(z) \sim z$ profiles for the nanoparticles in network cells with various $g$ (Fig. 3a and Supplementary Fig. S2), consolidating the general nature of these regimes. Previous studies focusing on elastic deformation of the local network loop also indicate the presence of Regime I[28,29]; by contrast, our theoretical approach allows detailed examination of the MEP, revealing the existence of Regimes II and III. Strikingly, by comparing the distributions of these regimes for different $g$, one can find that a larger $g$ gives rise to smaller areas of Regimes II and III, underscoring the correlation between scaling regimes and cell topologies.

Extending the analysis, we examine the free energy profiles at certain values of $r_{out}$, $r_{mid}$ and $r_{in}$. As schematically shown in Figs. 1b and 3b, $r_{out}$ can be considered the critical particle size causing deformation of the vertices, i.e., cross-links, of the network cells. Setting $R = r_{out}$, we obtain the boundary between Regimes I and II, as denoted by the magenta dashed curve. Likewise, $r_{mid}$ can be regarded as the critical size inducing deformation of the edges, i.e., strands. With $R = r_{mid}$, the boundary between Regimes II and III can be characterized by the black dashed curve. Thus, the boundaries dividing these different regimes can be determined based on $r_{in}$, $r_{mid}$, and $r_{out}$, as illustrated by the diagrams in Fig. 3b. As indicated by Eq. 1, with the increase of $g$, the aspheric parameters $r_{mid}/r_{in}$, $r_{out}/r_{in}$ decrease and are gradually approximate to 1.0, corresponding to an anisotropic-to-isotropic transition of the network cell[34]. Thus, the boundaries between $R = r_{out}$ and $r_{mid}$, and between $R = r_{mid}$ and $r_{in}$ approach to each other, giving rise to a shrinkage of Regimes II and III. In particular, for the network cell with very large $g$ where $r_{in}$, $r_{mid}$ and $r_{out}$ are approximately equal, these boundaries can be anticipated to superpose on each other, and then, Regimes II and III will disappear, reverting to the previous results focusing on isotropic deformation of a circular loop. This trend can also be identified from the free energy barrier experienced by a nanoparticle migrating from a cell to its neighboring cell, that is, $U_b$. In Fig. 3c, we show the plots of $U_b$ against $d$ for various $g$, and fitting $U_b$ at $d = 2r_{out}$ and $2r_{mid}$ in each plot leads to two boundaries (magenta and black dashed lines) separating the diagram into three characteristic regions corresponding to the scaling regimes in Fig. 2a, b. When $d > 2r_{out}$, the hopping energy barrier behaves quadratic dependence on $d$, consistent with the studies of a circular loop[28]. However, such a scaling behavior does not hold in other regimes. The dashed lines represent the theoretical prediction to $U_b(2r_{out})$ and $U_b(2r_{mid})$ for various cell topologies, giving the boundaries separating the characteristic regimes. The two boundaries of the regimes tend to be asymptotic with increasing $g$, in accordance with the above mentioned results.

**Topology-mediated dynamical regimes.** To further pinpoint the physical origin of the impact of network cell topology, we turn to the nanoparticle dynamics in these regimes as well as their dependence on the cell topology (see Methods). Based on the free energy landscape established above, we numerically obtain the nanoparticle diffusion coefficient $D$ in response to various values of $d$ and $g$ (Fig. 4 and Supplementary Fig. S3). By plotting the diffusion coefficient $D$ normalized by its value at $2r_{in}$ versus $d$ normalized by $2r_{in}$, the nanoparticle diffusivities at different $g$ collapse on a master curve and present a power-law slope of $-3$ when $d \leq 2r_{in}$, demonstrating that the diffusivity is

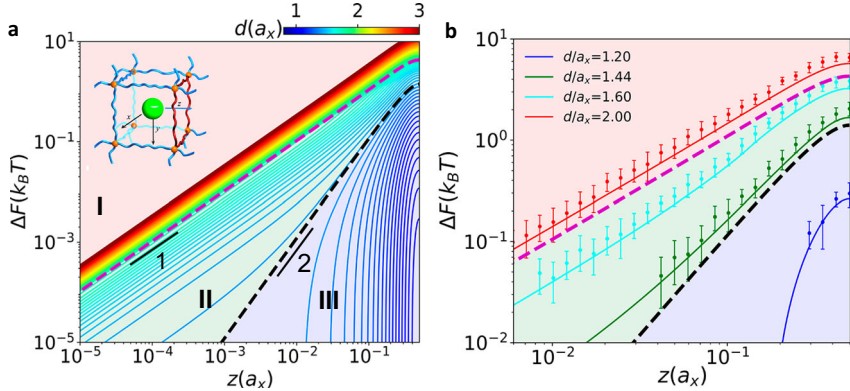

**Fig. 2 Distinct scaling regimes of free energy change. a** Dependence of $\Delta F(z)$ on the position $z$ for various $d$ in the log-log scale, obtained from the theoretical model of a network-particle system with $g = 6$. The upper left inset shows the axes of the system, where $z$ axis is along the mean energy path (MEP). The dashed lines represent the theoretical boundaries separating different regimes. **b** The comparison between MC (points) and theoretical results (lines) for some representative particle sizes. The error bar indicates the standard deviation.

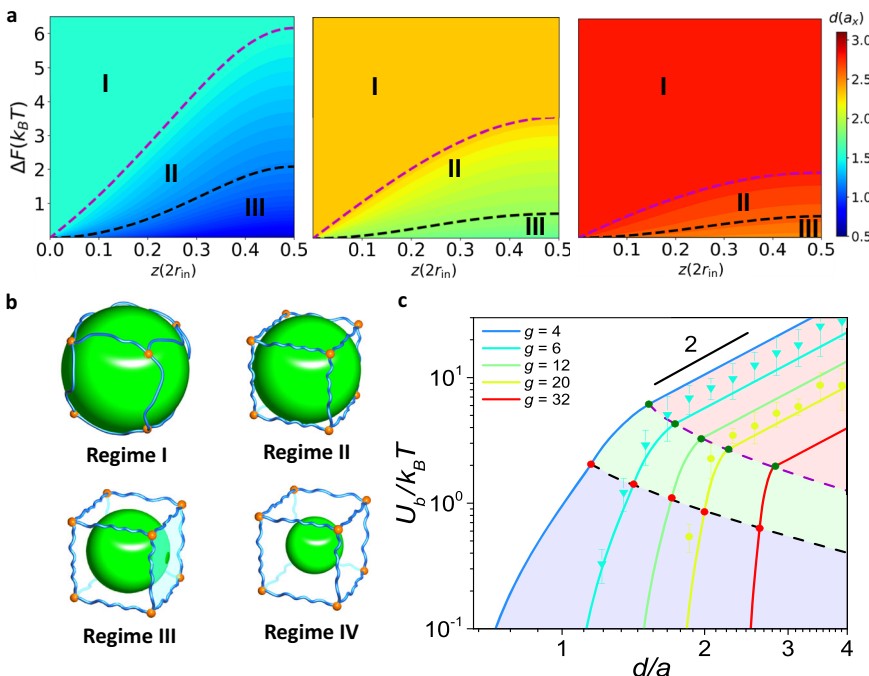

**Fig. 3 Topology-dictated scaling regimes of free energy change. a** Heat map of $\Delta F$-$z$ for various $d$ in network cells at $g = 4$ (left), 20 (middle) and 32 (right). The color bar indicates the values of the particle diameter $d$. The boundaries of Regimes I and II and Regimes II and III are represented by purple and black dashed lines, respectively. Specifically, the hidden Regime IV in each plot gives $\Delta F(z) = 0$. **b** Schematic of spherical particles with various sizes in a network cell for different regimes: Regime I: $r > r_{out}$, Regime II: $r_{mid} < r < r_{out}$, Regime III: $r_{in} < r < r_{mid}$, Regime IV: $r < r_{in}$. **c** Free energy barrier of the particle between neighboring cells $U_b$ against $d$ for various cell topologies. MC results are also plotted for topologies of $g = 6$ (blue), and 20 (yellow). The error bar indicates the standard deviation.

completely governed by the local Rouse dynamics[35] of the strands (Fig. 4a). However, the $D$ of intermediate size nanoparticles ($2r_{in} < d < 2r_{out}$) crosses over to an extreme decrease. When $d \geq 2r_{out}$, the large nanoparticle causes radial dilation of the cell vertices (Fig. 3b, Regime I), resulting in an almost isotropic deformation of the cell that resembles the deformation of a circular loop; consequently, the diffusivity reverts to the exponential dependence, which can be corroborated by the circular scatters in Fig. 4a, b as well as the collapsed curves of $D/D(2r_{out}) \sim d/(2r_{out})$ in Fig. 4b. Recall that the studies focusing on local deformation of a circular loop showed that the diffusion of small and large nanoparticles, divided by the size of $a_x$, exhibits the dynamical

regimes of $D \sim (d/a_x)^{-3}$ and $D \sim \exp(-d^2/a_x^2)$, respectively[28,35]. In contrast, the emergence of the intermediate regime and the change in the switching points from $a_x$ to $r_{in}$ and $r_{out}$ for different regimes in Fig. 4 highlight the impact of cell topology on the nanoparticle diffusion dynamics. For $d > 2r_{in}$, the nanoparticle starts to experience the free energy landscape on the faces of a network cell (Fig. 3b, Regime II), triggering the cell topology effect, which induces the deviation from the exponential dependence denoted by the colored dashed curves in Fig. 4a.

To provide a refined picture of the mechanism underpinning the topological effect on nanoparticle dynamics,

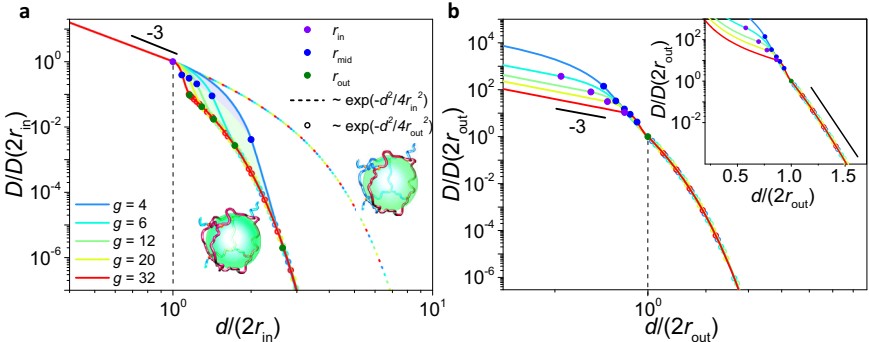

**Fig. 4 Dynamical regimes mediated by network topologies. a** $D/D(2r_{in})$ against $d/2r_{in}$ for various network cell topologies in the log-log scale. Inset: schematic diagrams of the deformation based on a network cell (left) and the deformation based on a network loop (right). **b** $D/D(2r_{out})$ against $d/2r_{out}$ for various network cell topologies in the log-log and the log-linear (inset) scales. The scaling parameters $r_{in}$ (purple), $r_{mid}$ (green) and $r_{out}$ (blue) are presented on each plot. Power law and exponential dependences on the ratio between $d$ and $2r_{in}$ (dashed line) or $2r_{out}$ (circular scatter) are depicted.

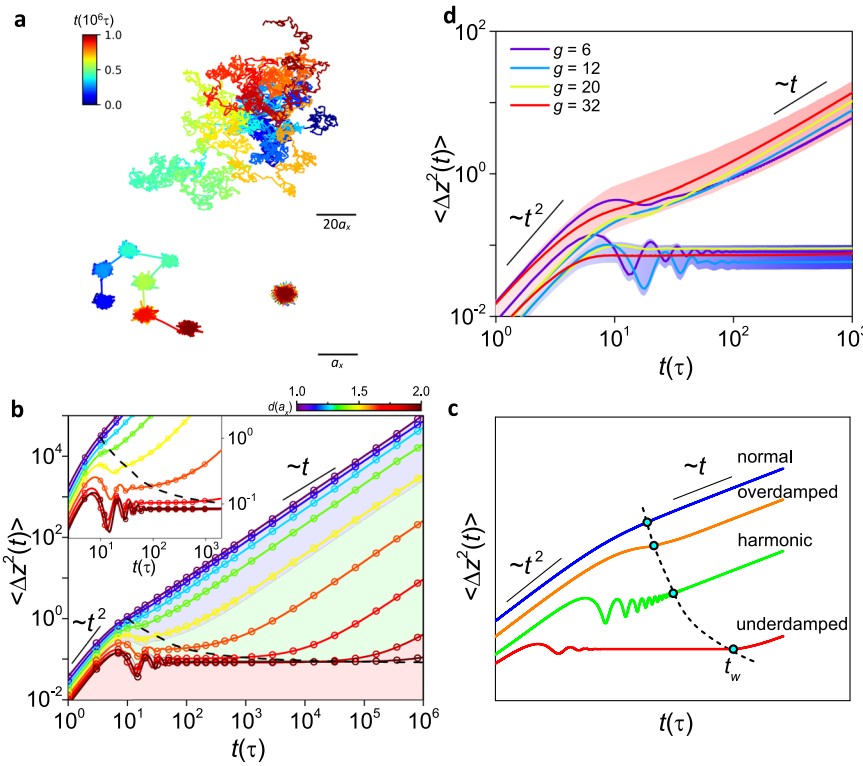

**Fig. 5 Topological effect on microscopic dynamics. a** Representative trajectories of Brownian (top), hopping (left) and trapped (right) dynamics in the macromolecular network at $g = 6$, where the diameter of particle $d/a_x = 1.00$, 1.65, and 1.90, respectively. Time scale is color-coded at the top-left corner, and length scales are shown on the right bottom of each plots. **b** $\langle \Delta z^2(t) \rangle$ as a function of $t$ for different $d/a_x$ at $g = 6$. The solid line denotes the theoretical results predicted by Eqs. 3, 5–7, and the hollowed scatters mark the results obtained from numerical simulations. The color of lines and scatters indicates the value of $d$ and is coded in the upper right panel. **c** Schematics of theoretical predictions of $\langle \Delta z^2(t) \rangle$ at Regimes I (red), II (green), III (orange) and IV(blue), where the oscillating state of each regime is labeled on the right of the plot. Circles denote the characteristic waiting time $t_w$, and a dashed line connecting circles is used to guide to the eye, as also illustrated by the black dashed lines in Fig. 5b and its inset. **d** $\langle \Delta z^2(t) \rangle$ as a function of $t$ for different topologies: $g = 6$ (purple), 12 (blue), 20 (yellow), 32 (red) at boundaries of Regimes II and III (upper, $d = 2r_{mid}$) and Regimes I and II (bottom, $d = 2r_{out}$), which are approximately grouped by the shaded sections.

we also develop theoretical models, complemented with simulations, to dissect the microscopic dynamics described by various parameters, such as the particle trajectory and mean square displacement (MSD) (see Supplementary Information III for more information). Figure 5a depicts three typical trajectories of nanoparticles undergoing Brownian, hopping and trapped dynamics, as confirmed by the MSD (Fig. 5b and Supplementary Fig. S4) and the spatial probability distribution function

$G_s(z, t)$ (Supplementary Figs. S5–S7), which resemble previous experimental[36,37] and theoretical[38,39] results. From the measured trajectories, we consider that the dynamical process of a nanoparticle confined in a network can be coarse-grained and decomposed into a series of consecutive jump and waiting events[40], i.e. $z(t) = \sum_{i=0}^{N(t)} \delta z_i(t)$, where $\delta z_i$ denotes the jump length of event $i$, and $N(t)$ is the counts of transition events. The number of jumps grows linearly with time on average, $\langle N(t) \rangle = t/t_w$,

where $t_w$ is the characteristic waiting time before escaping from a network cell (see Supplementary Fig. S8). Thus, the total MSD of the nanoparticle takes the forms,

$$\langle \Delta z^2(t)\rangle = \begin{cases} \frac{t}{t_w}\langle \delta z^2(t)\rangle & t \geq t_w \\ \langle \delta z^2(t)\rangle & t < t_w \end{cases} \quad (2)$$

which captures the trapped and hopping motions in the intermediate time scale and recovers back to the normal diffusion in a long-time limit (Fig. 5b).

**Oscillation modes orchestrated by topologies.** Strikingly, our simulation reveal that before escaping from a network cell, particles can exhibit various oscillation modes in different regimes (Fig. 5b). To elucidate the underlying relationship between $\Delta F$ and the oscillation modes, we analytically obtain underdamped, oscillated, and overdamped modes from Regime I to III, as demonstrated in Fig. 5b and schematically shown in Fig. 5c. We focus the dynamical process for the short time scale with $t < t_w$, where hopping events don't happen and it is unnecessary to distinguish between $\delta z$ and $z$. In this case, the microscopic dynamics of the nanoparticle in different regimes will significantly depend on the form of the energy landscape, discussed as follow (see Supplementary Information III for more details):

In Regime I, the potential $\Delta F(z) = \rho|z|$ and is "V-shaped", with $\rho = U_b/r_{in}$ being an arbitrary constant. The MSD in Laplace domain gives

$$\langle \Delta z^2(s)\rangle = \frac{8D_0^2}{\rho^2}\frac{1}{s\left(1 + \sqrt{1 + \frac{4D_0(1/t_w+s)}{\rho^2}}\right)^2} \quad (3)$$

Given that the characteristic equation $s(1 + \sqrt{1 + 4D_0(1/t_w + s)/\rho^2})^2 = 0$ has a complex root $s_1 = (-\rho^2/4D_0 - 1/t_w) + (\rho^2/4D_0)i$, the Laplace transformation of Eq. 3 has the form $\langle \Delta z^2(t)\rangle \sim e^{s_1 t}$ at intermediate time scale ($t \to 1/|s_1|$), following an underdamped mode of oscillation (see Supplementary Information IV for more information). At long time scale ($t \to \infty$), the MSD in Regime I saturates to a constant, given by

$$\langle \Delta z^2(t)\rangle \sim 2D_0 t_w - \rho^2 t_w^2(\Delta_0 - 1) \quad (4)$$

where $D_0 = k_B T/6\pi\gamma R$ is the diffusion constant of the nanoparticle in solvents and $\Delta_0 = \sqrt{1 + 4D_0/(\rho^2 t_w)}$.

In Regime II with $\Delta F(z) = \omega^2 z^2/2$, we proceed to apply Langevin's original strategy to study the problem of the harmonic oscillator Brownian motion, where $\omega = (2U_b)^{1/2}/r_{in}$ is the frequency of harmonic oscillator. It can be determined that for $\gamma > \sqrt{8}m\omega$, the MSD reads

$$\langle \Delta z^2(t)\rangle = \frac{k_B T}{m\omega^2}\left[1 - e^{-\frac{\gamma t}{2m}}\left(2\sinh^2\left(\frac{1}{4}\beta_1 t\right) + \frac{\gamma}{m\beta_1}\sinh\left(\frac{1}{2}\beta_1 t\right) + 1\right)\right] \quad (5)$$

where $\beta_1 = \sqrt{(\gamma/m)^2 - 8\omega^2}$, the oscillation mode is overdamped. For $\gamma < \sqrt{8}m\omega$, the MSD reads

$$\langle \Delta z^2(t)\rangle = \frac{k_B T}{m\omega^2}\left[1 - e^{-\frac{\gamma t}{2m}}\left(-2\sin^2\left(\frac{\sqrt{2}}{2}\omega_1 t\right) + \frac{\gamma\sin(\sqrt{2}\omega_1 t)}{\sqrt{8}m\omega_1} + 1\right)\right] \quad (6)$$

where $\omega_1 = \sqrt{\omega^2 - (\gamma/m)^2/8}$, the oscillation mode becomes underdamped.

In Regime III with $\Delta F \sim o(z^2)$, we can identify that the diffusion dynamics as the random walk with double reflecting boundaries at $z = -r_{in}$ and $z = r_{in}$. The MSD has the form[41]:

$$\langle \Delta z^2(t)\rangle = \frac{r_{in}^2}{6}\left[1 - \exp\left(-\frac{t}{\tau_0}\right)\right] \quad (7)$$

which indicates the overdamped mode of oscillation. In the long-time limit, it reduces to $\langle \Delta z(t)\rangle \sim r_{in}^2/6$. Furthermore, the theoretical predictions (solid lines) are well approximated by the numerical data (hollow circles). Given that the free energy landscape remarkably depends on the network cell topology, the oscillation modes of nanoparticles can be orchestrated by tailoring the cell topology, which is supported by the reduced oscillation in the MSD plots at the regime boundaries with increasing $g$ (Fig. 5d).

Oscillatory behavior in physical systems is a typical signature of inertial effects, reflecting competition between the force of the potential and the viscosity[42]. For a given system with a well-defined relaxation time, oscillations occur only for a certain form of the energy landscape (see Supplementary Information IV for more details). Briefly, in Regime I where a nanoparticle is confined in an entire cell, it is always subjected to negative response with a constant force $\rho$, inducing the oscillatory behavior. In Regime II, affected by partial chains of the cell, the nanoparticle turns to receiving negative response with a linear restoring force $\omega z$, which competes with the frictional force. When the restoring force is large, this corresponds to obvious oscillatory behavior. By contrast, when the restoring force becomes small, nanoparticle dynamics is essentially diffusive. In Regime III where the nanoparticle experiences the free energy at the boundary of the cell, the response reduces to zero. Therefore, the dynamics in Regime III is dominated only by the frictional force, and the oscillation is overdamped. However, as only a local network loop is considered in previous works[12,28,29], the viscosity force of the nanoparticle is far larger than the restoring force around $z = 0$; thus, oscillatory behavior of the nanoparticle is suppressed and the underdamped modes of oscillation cannot be observed.

## Discussion

Taken together, we develop theoretical approaches to provide a fundamental research of the free energy landscape and its dependence on cell topology for the transport of nanoparticle entrapped in macromolecular network. Our theoretical results, coupled with simulations, push the envelope of the full view of the free energy landscape sculpted by the network topology and thereby its impact on the transport of nanoparticles, leading to distinct scaling regimes regarding the free energy landscape and nanoparticle dynamics. The findings isolate topology as a key feature governing the dynamic behavior in networks, altering the conventional picture of the physical origin of nanoparticle transport in network environments.

In addition, the synthesis of macromolecular networks with well-controlled architectures is coming within reach[25–27], allowing facile approaches to tailor the network topology. While here we focus on the permanently cross-linked regular networks, we speculate that the theory can be ready to extend to the nanoparticle diffusion in unentangled and entangled macromolecular networks with irregular cells. The spherical nanoparticle can also be readily replaced by diverse anisotropic nanoparticles through coupling the shape factor into the theoretical approaches. Thus, the formulated theoretical approaches can serve as a foundation for further exploration of topological effects on the dynamic behavior in various networks, synthetic or biological. We believe that this work will certainly stimulate efforts into the above promising topics of interest to physicists and materials scientists.

## Methods

**Development of analytical model for free energy landscape.** To quantitatively examine how cell topology affects the free energy landscape of particles, we develop a new theoretical model to calculate the free energy experienced by a spherical particle in

networks. Full details can be found in Supplementary Information I. Briefly, we consider a hard spherical nanoparticle of radius $R$ in a cross-linked polymer network of Gaussian chains without dangling end. The network topology is specified by (i) the set of cross-links $k = \{\mathbf{r}_i\}_{i=1}^M$, with $M$ cross-links between the efficiently bridged Gaussian chains, (ii) the collection of linker connections marked as the tuple and (iii) the continue curve path of linked strands $\mathbf{r}_{ij}(s)$ with contour variable $s \in [0, 1]$. For a Gaussian chain of $N$ bonds of Kuhn length $b$, the contour length $L = Nb$, and the average mesh size, $a_x = N^{1/2}b = 1$, is the unit length of the system. Considering the cross-linked chains in a dilute solution of $\theta$-solvent, where screened excluded-volume statistics can be assumed, the interaction potential between monomers is ignored to recover the ideal statistics[43]. Hence, in the canonical ensemble, the Helmholtz free energy of the particle-network system is determined by $F(\mathbf{r}_{np}) = -k_B T \ln Z(\mathbf{r}_{np})$, where the partition function $Z(\mathbf{r}_{np})$ takes the form[44,45],

$$Z(\mathbf{r}_{np}) = \prod_k \int d\mathbf{r}_k \prod_{(i,j)} \int D\mathbf{r}_{ij}\delta(\mathbf{r}_{ij} - \mathbf{r}_i)\delta(\mathbf{r}_{ij} - \mathbf{r}_j) \times \exp[-\beta \sum_{(i,j)} H(\mathbf{r}_{ij}, \mathbf{r}_{np})] \quad (8)$$

$\mathbf{r}_k$ is the position vector of the cross-linked point $k$, $\mathbf{r}_{ij}(s)$ is the path vector of the strand with its start $\mathbf{r}_{ij}(0) = \mathbf{r}_i$ and end $\mathbf{r}_{ij}(1) = \mathbf{r}_j$, $\mathbf{r}_{np}$ is the position vector of the particle, $\beta = 1/k_B T$, and $\delta$ is the delta function. Coupling the excluded volume effect of a hard sphere with radius $R$, the modified Hamiltonian of the strand between cross-link pair $(i, j)$ is given by,

$$H(\mathbf{r}_{ij}, \mathbf{r}_{np}) = \frac{3k_B T}{2Nb^2} \int_0^1 ds \left\|\frac{\partial \mathbf{r}_{ij}}{\partial s}\right\|^2 \Phi(|\mathbf{r}_{ij} - \mathbf{r}_{np}| - R) \quad (9)$$

where $s \in [0, 1]$ is the contour variable, $N$ is the number of bonds in a strand, $b$ is the Kuhn length, and $\Phi(x)$ is the unit step function.

**Numerical simulation by nonlinear Langevin equation (NLE).** Establishing the free energy landscape for a nanoparticle in a network cell with defined topology allows us to examine the nanoparticle dynamics through NLE which has been successfully applied to study the diffusion dynamics of colloids or polymers[35,46,47]. Considering the motion of a nanoparticle along the $z$ axis, the NLE equation can be written as

$$m\frac{d^2z}{dt^2} = -\Delta F'(z) - \gamma\frac{dz}{dt} + \xi(t) \quad (10)$$

where $\Delta F(z)$ is the free energy change experienced by the nanoparticle at position $z$, $m$ is the nanoparticle mass, $\gamma$ is the friction coefficient, $\xi(t)$ represents the fluctuating force satisfying $\langle\xi(t)\rangle = 0$, $\langle\xi(t)\xi(t')\rangle = 2\gamma k_B T\delta(t - t')$, and $\langle...\rangle$ stands for ensemble average. For the numerical simulations, the Euler-Maruyama (EM) method[48] is applied to solve the stochastic differential equation (SDE) of Eq. 10, where the integrate time step $\Delta t = 0.01\tau$, $\gamma/m = 10.0\tau^{-1}$ and $\tau$ is the unit of the time. Diffusion coefficient $D$ of a nanoparticle can thereby be obtained through the correlation function, $D = \int_0^{t_c}\langle v(t) \cdot v(0)\rangle dt$, where $v(t) = dz/dt$ represents the velocity of the nanoparticle, and $t_c = 10^7\tau$ is the total simulation time.

**Monte Carlo (MC) simulation.** In the MC simulations, we use the Wang-Landau[49,50] method to accelerate the extraction of free energy in respect to $z$, $F(z)$. The flat histogram or Wang-Landau sampling method, where an automatically generated bias or penalty function, $f(z)$, is applied to the system along $z$ coordinates, so that the configurational integral reads

$$Z(z) = \exp(-\beta z)\int d\{\mathbf{r}\}\exp[-\beta H(\{\mathbf{r}\}, z)] \quad (11)$$

where $\{\mathbf{r}\}$ denotes configurational space at a given state $X$. For every visit to a state along the coordinate, a small penalty energy, $f_0 = 0.5$, is added to $f(z)$ until $Z$ is equal for all $X$. Thus, during simulation the free energy landscape is flattened, while the true free energy is simply the negative of the generated bias function

$$\beta F(z) = -\ln \int d\{\mathbf{r}\}\exp[-\beta H(\{\mathbf{r}\}, z)] \quad (12)$$

In our simulations, at least fifty independent runs are performed for each parameter set, so that the standard error is estimated within $0.3k_B T$. The detailed derivation of the Hamiltonian $H(\{\mathbf{r}\}, z)$ can be found in Supplementary Information V.

## Data availability

The data supporting the findings of this work are available within the paper and the Supplementary Information files. Source data are provided with this paper.

## Code availability

The code developed for this paper is made available at https://doi.org/10.5281/zenodo.6794578.

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

## Acknowledgements

The authors acknowledge support from the National Science Foundation of China (Grant Nos. 22025302, 21873053, 51633003), the National Key R&D Program of China (Grant No. 2016YFA0202500). We thank Pengyu Chen for stimulating discussion on the simulation results.

## Author contributions

L.T.Y. designed the research and conceived the project. X.D., X.Z. and L.T.Y. contributed to the development of the computational model and to its interpretation. X.D. and L.T.Y. developed the analytical models and interpreted the results. L.T.Y., X.D., L.G. and Z.X. wrote the paper. All authors discussed the results and commented on the manuscript.

## Competing interests

The authors declare no competing interests.
