## [Peer Review File · Nature Communications]

REVIEWER COMMENTS

Reviewer #1 (Remarks to the Author):

This is a very study of nanoparticle diffusion in networks of various topology. The results are noteworthy and worthy of support. Overall a very important contribution to the field and a great guide for experimentalists to interpret current experiments (ref 35, 36) and optimize new experiments. I do have several comments.

1) The paper is exquisitely written. However, the results section does get into the weeds a bit making it hard for non experiments. The introduction and discussion are quite clear.

2) To be of more use, it would be helpful to discuss the results in terms of the distance travelled by the nanoparticles. The MSD data in Figure 5 for example shows trapped particles that explore a small region within the cage. These values are within the data in 5b and d. This would be valuable b/c it tells us the effective size of the fluctuating network.

3) Similarly , when discussing the activation energy in Figure 3c, values greater than say 2-3kbT are likely inaccessible. So only the $d=2r_{mid}$ data is relevant to experimental observation.

4) Figure 3a shows how g impacts regimes I, II and III. More discussion of mechanism underlying this change would help the reader.

5) Fig 5b and d nicely capture experiments in 36. Very nice.

Reviewer #2 (Remarks to the Author):

The authors performed theoretical analyses and computational simulations to investigate the effect of network topology on the transport of nanoparticles in macromolecular networks. The authors obtained the free energy landscape experienced by the nanoparticles, and illustrated that the network topology has impact on the free energy landscape and the dynamics of the nanoparticles. They showed that the radius of the circumscribed sphere of the topology network is a characteristic scale for nanoparticles, beyond which the diffusion coefficient of nanoparticles decreases exponentially. The results provide new and interesting physical insights on the transport of nanoparticles in macromolecular networks. I have some comments need the authors to address before the acceptance of the paper.

(1) The authors showed that r_{out} , the radius of the circumscribed sphere of the topology network, rather than a_x , the mesh size, is the characteristic scale for nanoparticles, beyond which the diffusion coefficient of nanoparticles decreases exponentially. In the theoretical model (ref 28 in the manuscript)

developed by Rubinstein et al., a_x is a characteristic scale that is comparable to the mesh size, which is not necessarily exactly the mesh size. In this case, r_{out} is comparable to a_x and the exponential decrease of the diffusivity should be similar.

(2) In Fig. 3c, the authors showed the energy barriers of nanoparticles for different network topologies in which with the same a_x , the area of the faces of these polyhedra is not the same. I feel it is also interesting/important to compare the energy barriers of nanoparticles when the area of the faces of the polyhedra is the same/similar and that when the circumradius of the faces of the polyhedra is the same/similar.

(3) In Fig. 3c, the Monte Carlo results always overestimate the energy barrier of nanoparticles, what is the origin of the difference?

(4) Please clarify the definition of the mesh size a_x for different polyhedral. Is it the mean end-to-end distance of the strands that constitute the edge?

(5) On page 8, the authors claimed “The two boundaries of the regimes tend to be asymptotic with increasing g , in accordance with the above mentioned results.” I feel the two dashed lines in Fig. 3c are not asymptotic, and the gaps between the two dashed lines are almost a constant for $g \geq 12$.

(6) I tried to check the correctness of Eq (1) in main text. For a regular hexahedron, $n=4$ and $g=6$. From Eq (1), we have $r_{in}/a_x = \sqrt{3}/6$; $r_{mid}/a_x = \sqrt{2}/2$; $r_{out}/a_x = \sqrt{3}/2$, then $r_{in} : r_{mid} : r_{out} = \sqrt{3}/3 : \sqrt{2} : \sqrt{3}$, which is different from the correct relationship, i.e., $r_{in} : r_{mid} : r_{out} = 1 : \sqrt{2} : \sqrt{3}$. Please check it.

(7) On page 3, the authors mentioned the following sentence “purification in porous materials [16]”. However, ref 16 is about the mechanics of cross-linked networks. Please update the reference. On page 8, the authors mentioned “the hopping energy barrier behaves quadratic dependence on d , consistent with the studies of a circular loop [18].” Ref 18 is about the pore size of mucus rather than the relationship between energy barrier and d . Please update the reference.

First of all, we would like to thank the reviewers for their thoughtful comments on our original manuscript. Our detailed responses to them are as follow:

Responses to Reviewer 1:

This is a very study of nanoparticle diffusion in networks of various topology. The results are noteworthy and worthy of support. Overall a very important contribution to the field and a great guide for experimentalists to interpret current experiments (ref 35, 36) and optimize new experiments. I do have several comments.

Reply: Thank you very much for the expert comments and kind suggestions!

1. The paper is exquisitely written. However, the results section does get into the weeds a bit making it hard for non-experiments. The introduction and discussion are quite clear.

Reply: Thanks for the comment. Indeed, in the sections of Introduction and Discussion, we detail the background and summarize the findings of this research, which, we believe, is essential for understanding the topological effects on the particle transport in macromolecular networks. In the section of Results, to elucidate physical origin behind the topological effects, we develop the theoretical model of a nanoparticle in macromolecular network and carry out necessary simulations. The topology-dictated scaling regimes of free energy landscape, diffusion coefficients, and oscillation modes are thereby discussed to realize fundamental analysis. Actually, the physical picture of the issue concerned in this work is general for both theoretical and experimental researchers. With comparison, our results do substantiate the experimental observations^{1,2} of the diffusion behaviors of nanoparticle in the networks. Moreover, the theoretical analysis brings new insight into the physics of the transport dynamics in confined media of networks.

2. To be of more use, it would be helpful to discuss the results in terms of the distance travelled by the nanoparticles.

Reply: Thanks for the comment. It does be important to test the distance travelled by the nanoparticles when considering their dynamics. Actually, in statistical mechanics, the mean squared displacement (MSD) is such a parameter measuring the deviation of the distance between the position of a particle and a reference position over time³. More importantly, through examining the scaling behavior exhibited by this parameter, various dynamics mechanisms can be further determined.

The MSD data in Figure 5 for example shows trapped particles that explore a small region within the cage. These values are within the data in 5b and d. This would be valuable b/c it tells us the effective size of the fluctuating network.

Reply: The expert comment is appreciated. The trapped state is the dynamic mode for the particle with diameter much larger than the mesh size of the network, where the MSD approaches a plateau related to its fluctuation. As rightly pointed out by the Reviewer, it provides an effective approach to obtaining the effective size of the fluctuating network, as demonstrated in Refs. 4 and 5.

3. Similarly, when discussing the activation energy in Figure 3c, values greater than say $2-3k_B T$ are likely inaccessible. So only the $d=2r_{mid}$ data is relevant to experimental observation.

Reply: Thanks for the expert comment. Physically, the activation energy U_b measures the probability for a nanoparticle escaping from a network cell⁶, which governs the diffusion behaviors of a nanoparticle in the networks with different topologies. Specifically, when the activation energy is lower than the thermal energy, i.e., $U_b < k_B T$, the nanoparticle behaves as Brownian dynamics. When the activation energy is too large, the escaping process is fully inhibited and thereby the nanoparticle is trapped in the network cell. Within the broad region in between, the hopping dynamics emerges. As demonstrated in previous theoretical⁷ and experimental⁸ works, the typical range of such an intermediate region of hopping dynamics is about $2.2k_B T < U_b < 9.0k_B T$. As shown in Figure 3c in the manuscript, even $U_b(2r_{out})$ is lower than $6.5k_B T$ for all cell topologies, which is not strong enough to restrict the nanoparticle

within a network cell. Thus, we believe that the diffusion behavior of nanoparticle discussed in the topology-dictated range ($2r_{in} < d < 2r_{out}$) is accessible and relevant to the experimental observation.

4. Figure 3a shows how g impacts regimes I, II and III. More discussion of mechanism underlying this change would help the reader.

Reply: Thank for the suggestion. As indicated by Eq. 1, with the increase of g , the aspheric parameters r_{mid}/r_{in} , r_{out}/r_{in} decrease and are gradually approximate to 1.0, corresponding to an anisotropic-to-isotropic transition of the network cell⁹. Thus, the boundaries between $R = r_{out}$ and r_{mid} , and between $R = r_{mid}$ and r_{in} approach to each other, giving rise to a shrinkage of Regimes II and III. In particular, for the network cell with very large g where r_{in} , r_{mid} and r_{out} are approximately equal, these boundaries can be anticipated to superpose on each other, and then, Regimes II and III tend to disappear. In this case, the network cell becomes almost isotropic, similar to the studies of a circular loop¹⁰.

In this revision, we have added more discussion regarding this aspect, which can be found from the 9th line to the 16th line of Page 8 (marked in red).

5. Fig 5b and d nicely capture experiments in 36. Very nice.

Reply: Thank you very much for the comment! Indeed, a reasonable agreement between the theoretical analysis and the important observation of this valuable experimental work can be identified, consolidating that our theoretical model captures the physics underlying the dynamics process of the nanoparticle diffusion in macromolecular network.

Responses to Reviewer 2:

The authors performed theoretical analyses and computational simulations to investigate the effect of network topology on the transport of nanoparticles in macromolecular networks. The authors obtained the free energy landscape

experienced by the nanoparticles, and illustrated that the network topology has impact on the free energy landscape and the dynamics of the nanoparticles. They showed that the radius of the circumscribed sphere of the topology network is a characteristic scale for nanoparticles, beyond which the diffusion coefficient of nanoparticles decreases exponentially. The results provide new and interesting physical insights on the transport of nanoparticles in macromolecular networks. I have some comments need the authors to address before the acceptance of the paper.

Reply: Thank you very much for the expert comments and kind suggestions!

(1) The authors showed that r_{out} , the radius of the circumscribed sphere of the topology network, rather than a_x , the mesh size, is the characteristic scale for nanoparticles, beyond which the diffusion coefficient of nanoparticles decreases exponentially. In the theoretical model (ref 28 in the manuscript) developed by Rubinstein et al., a_x is a characteristic scale that is comparable to the mesh size, which is not necessarily exactly the mesh size. In this case, r_{out} is comparable to a_x and the exponential decrease of the diffusivity should be similar.

Reply: Thanks for the comment. In Ref. 10 (i.e., Ref. 28 in the main text), the theoretical model is derived based on the elastic deformation of a single network loop, where the parameter of a_x is considered as the characteristic scale. However, through examining the free energy landscape and the detailed dynamics, our work clarifies that the topology of the network cell, consisting of a group of loops, has a profound effect, where three parameters, i.e., r_{in} , r_{mid} , and r_{out} , which characterize the cell topology (see Eq.1), should be included as the characteristic scales. Indeed, these characteristic scales divide the new scaling regimes related to the topology, as shown in Figs. 2, 3a and 4.

To further confirm the physical meaning of a_x , we survey the literatures more carefully and ensure that it does represent the mesh size, as defined in the Abstract of Ref.10, reads “...the mesh size a_x of unentangled polymer networks...” and stated, for example, in the 4th paragraph of the left column on Page 2 in the same literature, reads “The network mesh size is defined as the average distance between two neighboring permanent cross-links along the chain...”.

(2) In Fig. 3c, the authors showed the energy barriers of nanoparticles for different network topologies in which with the same a_x , the area of the faces of these polyhedra is not the same. I feel it is also interesting/important to compare the energy barriers of nanoparticles when the area of the faces of the polyhedra is the same/similar and that when the circumradius of the faces of the polyhedra is the same/similar.

Reply: The expert comment is highly appreciated. Actually, as aforementioned in our reply to Comment 1, the characteristic scales are r_{in} , r_{mid} , and r_{out} instead of a_x when considering the cell topology of a network. To examine the effect of the area and circumradius of the faces, here we still perform additional calculation of the energy barrier U_b for the network cells with the same face, as detailed below:

In our study, the circumradius r_{circum} and area A of a face can be written as,

$$r_{circum} = \frac{a_x}{2 \sin(\pi / n)} \quad (1)$$

$$A = \frac{a_x^2}{2} \sin\left(\frac{2\pi n}{n+2}\right) \quad (2)$$

where a_x is the length of the edge of the face, and n the number of the edges of the face. Eqs.1 and 2 indicate that when two network cells have the same a_x and n , the circumradius r_{circum} and area A of their faces should be the same.

Figure R1. The energy barrier U_b as a function of $d/(2r_{circum})$ for different topologies $g = 4$ (red) and 20 (blue), where r_{circum} and A of the network cells have the same values.

Thus, we calculate U_b of the network cells with genus $g = 4$ and 20 , because they have the same a_x ($a_x = 1$) and n ($n = 3$), and thereby the same circumradius ($r_{\text{circum}} = \sqrt{3}a_x / 3$) and area ($A = \sqrt{3}a_x^2 / 4$). Figure R1 shows the free energy barrier U_b as a function of d/r_{circum} for these both network cells. It can be found that the energy barriers of the network cells are not the same for different topologies, in stark contrast to the result based on a single loop^{10,11}. However, the plots of U_b exhibit the similar trends to those in Fig.3c, demonstrating that the circumradius and area of the face do not change the scaling regimes dictated by the topology.

(3) In Fig. 3c, the Monte Carlo results always overestimate the energy barrier of nanoparticles, what is the origin of the difference?

Reply: Thanks for the careful comment. In the theoretical analysis, ΔF is calculated along the minimum energy path (MEP) from a cell to its neighboring cell. However, in Monte Carlo (MC) simulation, ΔF denotes the average over all of the allowed paths, leading to a lightly positive error in comparison with the theoretical value. Nevertheless, one can find that it has little impact on the scaling dependence of ΔF on z .

(4) Please clarify the definition of the mesh size a_x for different polyhedral. Is it the mean end-to-end distance of the strands that constitute the edge?

Reply: As illustrated in Figure 1b, a_x is defined as root-mean-square end-to-end distance of the strands that constitute the edge. For a Gaussian chain, $a_x = N^{1/2}b$, where N denotes bonds in the strand and b is the Kuhn length. This definition is the same as that in the previous work¹⁰.

In this revision, we have added the description regarding this aspect in the manuscript, which can be found from the 9th to 10th line of Page 5.

(5) On page 8, the authors claimed “The two boundaries of the regimes tend to be asymptotic with increasing g , in accordance with the above mentioned results.” I feel the two dashed lines in Fig. 3c are not asymptotic, and the gaps between the two dashed lines are almost a constant for $g \geq 12$.

Reply: Thanks for the comment. In Figure 3c, to determine the scaling laws, the dependence of U_b on d for various g is plot in the log-log scale. Here, to present a more definite relation between the two boundaries of the regimes, we plot the same data in the linear-linear scale. As shown in Figure R2, the two dashed lines of the boundaries do tend to be asymptotic with the increase of g .

Fig. R2 U_b against d for various cell topologies in the linear-linear scale.

(6) I tried to check the correctness of Eq (1) in main text. For a regular hexahedron, $n=4$ and $g=6$. From Eq (1), we have $r_{in}/a_x = \sqrt{3}/6$; $r_{mid}/a_x = \sqrt{2}/2$; $r_{out}/a_x = \sqrt{3}/2$, then $r_{in} : r_{mid} : r_{out} = \sqrt{3}/3 : \sqrt{2} : \sqrt{3}$, which is different from the correct relationship, i.e., $r_{in} : r_{mid} : r_{out} = 1 : \sqrt{2} : \sqrt{3}$. Please check it.

Reply: The comment is appreciated. After checking the forms in Eq.1 more carefully, we find that there does be typesetting errors in it. We are sorry for the errors although they do not influence other results and conclusions. Here we provide the corrected forms as follow:

$$\begin{aligned}
 r_{in} &= \frac{a_x}{2} \cot\left(\frac{\pi}{n}\right) \tan\left(\frac{\theta}{2}\right) \\
 r_{mid} &= \frac{a_x}{2} \cot\left(\frac{\pi}{n}\right) \sec\left(\frac{\theta}{2}\right) \\
 r_{out} &= \frac{a_x}{2} \cot\left(\frac{\pi}{n}\right) \sqrt{\sec^2\left(\frac{\theta}{2}\right) + \tan^2\left(\frac{\pi}{n}\right)} \quad (3)
 \end{aligned}$$

where θ represents the dihedral angle between any two faces. For platonic polyhedra, it can be written as,

$$\tan \frac{\theta}{2} = \frac{\cos(\pi / k)}{\sin(\pi / h)} \quad (4)$$

where h denotes the Coxeter number of the polyhedra, and $h(h+2)=2gn$.

In particular, for a regular hexahedron, $n = 4$, $k = 3$ and $g=6$. From Eq.3, one can calculate that $h = 6$, $\tan \frac{\theta}{2}=1$, $\sec \frac{\theta}{2}=\sqrt{2}$, and $\cot \frac{\pi}{n}=1$, such that

$$r_{\text{in}} = \frac{a_x}{2}, r_{\text{mid}} = \frac{\sqrt{2}}{2} a_x, r_{\text{out}} = \frac{\sqrt{3}}{2}.$$

In this revision, we have revised the Eq.1 and the description regarding it, which can be found from the 18th to 21st line of Page 5 (mark in red).

(7) On page 3, the authors mentioned the following sentence “purification in porous materials [16]”. However, ref 16 is about the mechanics of cross-linked networks. Please update the reference. On page 8, the authors mentioned “the hopping energy barrier behaves quadratic dependence on d, consistent with the studies of a circular loop [18].” Ref 18 is about the pore size of mucus rather than the relationship between energy barrier and d. Please update the reference.

Reply: Thank you very much for the careful comment! We check all the reference in the manuscript carefully and update the relevant literature. In particular, the reference regarding “purification in porous materials”, i. e., Ref. 16, is replaced by a more exact literature. The reference regarding “with the studies of a circular loop” should be Ref. 10 (i.e., Ref. 28 in the manuscript).

In this revision, we have updated the reference and its label regarding this aspect, which can be found for Refs. 16 and 28 (mark in red).

With these statements, we sincerely hope that the paper is now suitable for publication.

Reference

1. Yu, M. et al. Rapid transport of deformation-tuned nanoparticles across biological hydrogels and cellular barriers. *Nat. Commun.* **9**, 2607 (2018).
2. Parrish, E., Caporizzo, M. A. & Composto, R. J. Network confinement and heterogeneity slows nanoparticle diffusion in polymer gels. *J. Chem. Phys.* **146**, 203318 (2017).
3. Hansen, J. & McDonald, I. *Theory of Simple Liquids*. (Elsevier Academic Press, 2006).
4. Jiang, L., Xie, Q., Tsang, B. & Granick, S. Single-crosslink microscopy in a biopolymer network dissects local elasticity from molecular fluctuations. *Nat. Commun.* **10**, 3314 (2019).
5. Jiang, L., & Granick, S. Real-space, in situ maps of hydrogel pores. *ACS Nano*, **11**, 204-212 (2017).
6. Hänggi, P., Talkner, P., & Borkovec, M. Reaction-rate theory: fifty years after Kramers. *Rev. Mod. Phys.*, **62**, 251 (1990).
7. Xu, Z. et al. Enhanced heterogeneous diffusion of nanoparticles in semiflexible networks. *ACS Nano* **15**, 4608-4616 (2021).
8. Wong, I. Y. et al, Anomalous diffusion probes microstructure dynamics of entangled f-actin networks, *Phys. Rev. Lett.* **92**, 178101 (2004).
9. Torquato, S. & Jiao, Y. Dense packings of the Platonic and Archimedean solids. *Nature* **460**, 876-879 (2009).
10. Cai, L. H., Panyukov, S. & Rubinstein, M. Hopping diffusion of nanoparticles in polymer matrices. *Macromolecules* **48**, 847-862 (2015).
11. Dell, Z. E. & Schweizer, K. S. Theory of localization and activated hopping of nanoparticles in cross-linked networks and entangled polymer melts. *Macromolecules* **47**, 405-414 (2014).

REVIEWERS' COMMENTS

Reviewer #1 (Remarks to the Author):

The revision is satisfactory.

Reviewer #2 (Remarks to the Author):

The authors have fully addressed my raised concerns and now this paper is acceptable.

First of all, we would like to thank the reviewers for their thoughtful comments on our original manuscript. Our detailed responses to them are as follow:

Responses to Reviewer 1:

The revision is satisfactory.

Reply: Thank you very much for the expert comments and kind suggestions!

Responses to Reviewer 2:

The authors have fully addressed my raised concerns and now this paper is acceptable.

Reply: Thank you very much for the expert comments and kind suggestions!

We sincerely hope that the paper is now suitable for publication.